# A universal *O*-glycosylation platform enabled by pyridinium catalysis using gas-releasing oxazolidinone-based carbamates donors

Xiaoting Qin[1], Lin Ke[2], Qinbo Jiao[2], Wenxu Zhen[2], Wentao Lin[2], Jiaxin Luo[2], Wenyang Chen[3], Tingbo Liu[4], Shiping Wang[1] & Chunfa Xu [2] ✉

Chemical glycosylation facilitates the scalable synthesis of structurally well-defined carbohydrates for functional studies and therapeutic development, with progress being driven by donors and strategies. Herein, we present an efficient and versatile *O*-glycosylation method utilizing newly designed, bench-stable and readily accessible oxazolidinone-based glycosyl carbamates as donors. This reaction is catalyzed by 2-pentafluorophenyl pyridinium salts under mild conditions and demonstrates elegant performance compared to conventional promoters. This robust protocol facilitates orthogonal, iterative and latent-active glycosylations for streamlined synthesis of oligosaccharides. Mechanistic studies, including NMR, deuteration and kinetic isotope effect experiments, establish that the pyridinium catalyst initiates glycosylation by first binding glycosyl acceptors. It then activates carbamate donors via $CO_2$ release to generate the oxocarbenium ion, which is identified as the rate-determining step of the glycosylation process.

Carbohydrates are essential for diverse biological processes[1], yet their structural complexity renders isolation from natural sources challenging. Chemical synthesis provides an efficient, scalable approach[2,3], facilitating therapeutic development and mechanistic studies. Recent advances, including automated chemical synthesis[4,5], pre-activation methods[6], orthogonal glycosylation[7], latent-active strategies[8], and iterative approaches[9], have streamlined glycan assembly by minimizing intermediate purification. However, their efficacy relies on glycosyl donors and glycosylation protocols, underscoring the need for innovative donor design and synthetic methodologies.

An ideal glycosyl donor requires an optimal stability-reactivity balance, facile accessibility, and inert leaving groups to ensure selective glycosylation. Classical donors like glycosyl halides[10] and acetimidates[11,12] remain fundamental, but advancements include sulfur[13,14], phosphorus[15,16], epoxy[17,18], ether[19,20], and particularly ester-based[21–27] alternatives. Enhanced reactivity has been achieved through isomerization, cyclization, precipitation, and strain-release strategies

(Fig. 1a). However, practical implementation is hindered by the lack of cost-effective, mild synthetic protocols with stable precursors and the non-recyclable nature of leaving groups, which generate inactive waste, adversely impacting atomic economy.

Gas-release activation represents a promising strategy for driving chemical reactions[28], however, its application in carbohydrate chemistry remains underdeveloped[14]. Glycosyl carbamates[29–34] are a unique donor class of donors that can be synthesized under mild conditions, and activated by $CO_2$ release to drive glycosylation. Since Ley's seminal report[29] on imidazole-based donors, this class has been used for *O*-glycoside synthesis. However, their inherent instability necessitates fresh preparation and reliance on stoichiometric $ZnBr_2$ for promotion. Notably, imidazole exchange side reactions compromise glycosylation efficiency[35]. Subsequent developments included Kunz's alkene-functionalized donor[30], which, although activated by *N*-iodosuccinimide, undergoes cyclization instead of gas release due to the poor leaving ability of the allylamine group. Kiessling's sulfonyl

[1]College of Chemical Engineering, Fuzhou University, Fuzhou, China. [2]Key Laboratory of Advanced Carbon-Based Functional Materials (Fujian Province University), College of Chemistry, Fuzhou University, Fuzhou, China. [3]Central Laboratory, Fujian Medical University Union Hospital, Fuzhou, China. [4]Department of Hematology, Fujian Medical University Union Hospital, Fujian Institute of Hematology, Fujian Provincial Key Laboratory on Hematology, Fuzhou, China. ✉e-mail: xucf@fzu.edu.cn

**Fig. 1 | The art of the project. a** Strategies for improving donor reactivity; **b** Development of oxazolidinone-based glycosyl donors for glycosylation with newly designed catalyst.

carbamate donor[31] improved stability and reactivity but still required stoichiometric TMSOTf for high stereoselectivity. Redlich's trichloroacetamide carbamate[32] demonstrated enhanced reactivity, while its practical application was hampered by decomposition during purification[33]. Notably, both sulfonamide and trichloroacetamide leaving groups compete in glycosylation[36,37], potentially disrupting reaction pathways. Despite these advances, carbamate donors still rely on conventional promoters, compromising their orthogonality with other synthetic strategies. Additionally, the stereoselective catalytic glycosylation employing glycosyl carbamates via gas-releasing remains in its infancy.

To improve the utility of glycosyl carbamates, we developed oxazolidinone-based glycosyl carbamates as donors, leveraging their commercial availability, good stability, and suppressed nucleophilicity[38] that minimizes side reactions. We established an efficient O-glycosylation system combining these carbamates with a 2-pentafluorophenyl pyridinium catalyst (Fig. 1b), offering distinctive advantages: (1) good donor stability and straightforward synthesis; (2) broad compatibility; (3) activation under mild conditions via a unique catalytic mechanism; (4) recyclable oxazolidinone leaving groups improving atomic economy; (5) orthogonal activation enabling seamless integration with existing methods.

## Results

### Development of oxazolidinone-based glycosyl carbamate

The oxazolidinone-based glycosyl carbamates were synthesized by reacting hemiacetal **S1** with acyl chloride **S2** (prepared in one step from oxazolidinone with triphosgene[39]) in the presence of DIPEA. With this method, diverse donors (**1a-1k**) encompassing various sugar types and protecting groups were successfully prepared (Fig. 2). Notably, the donor exhibited good stability, maintaining integrity for >3 months at room temperature (Supplementary Fig S27).

### Reaction development

The reactivity of these newly designed donors in glycosylation was investigated using donor **1a** and 4-fluorophenol **2a** as model substrates under varying conditions (Table 1). Initial screening of common promoters (TMSOTf, TfOH, TsOH, BF$_3$·Et$_2$O, SnCl$_4$) revealed minimal reactivity (entry 1) or uncontrolled selectivity (entries 2–5). Phosphoric acid **A** marginally improved selectivity but suffered from low yield (entry 6). Pyridinium catalysts[40,41] were then explored; while **B** failed to initiate the reaction (entry 7), introducing a phenyl group at the *ortho* position did not enhance reactivity (entry 8). Substitution with a strongly electron-withdrawing pentafluorophenyl group significantly improved performance, yielding the product **3a** in 71% yield with excellent β-selectivity (entry 9). However, a bromide counterion

## General route for synthesis of oxazolidinone-based glycosyl carbamate

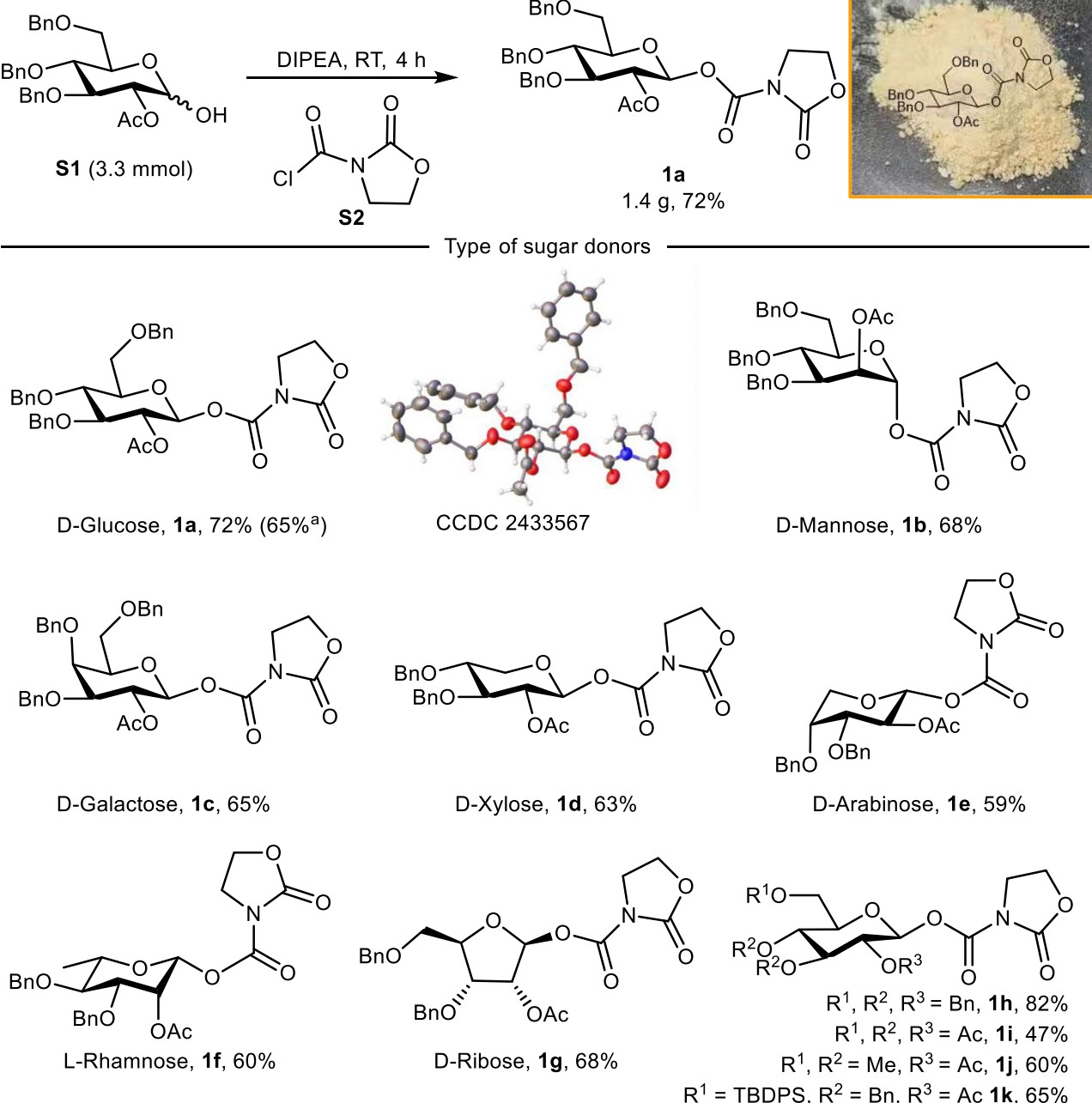

**Fig. 2 | Synthesis of oxazolidinone-based glycosyl carbamates.** Isolated yields. Bn = benzyl; Ac = acetyl; Me = methyl; TBDPS = *tert*-butylphenylsilyl Reaction condition: **S1** (3.3 mmol), **S2** (3.3 mmol), DIPEA (2.0 equiv.) RT, 4 h. [a]**S1** (6.2 g, 12.6 mmol) was used.

reduced both yield and selectivity (entry 10). Solvent screening showed that high-polarity solvents inhibited the reaction (entry 11). Pleasingly, increasing the stoichiometry of **1a** to 1.5 equiv, and using only 5 mol% catalyst **D** improved the yield to 96% with recovering 93% of the oxazolidinone (entry 12, Supplementary Fig S12). Surprisingly, when **1a-α** was employed as the donor, the reaction proceeded very slowly, even with a catalyst loading of 20% (entry 13). This observation suggests the 2-OAc group in the β-anomer may play a crucial role in facilitating the departure of anomeric leaving group[42]. Further studies on alternative donor **1h** confirmed that neighboring group participation was essential for high selectivity and reactivity (entry 14). Notably, the disarmed peracetylated donor **1i** proved completely unreactive (entry15). The failure of glycosylation with **1l**[32] underscores the good

reactivity profile of our donor (entry 16). Furthermore, control experiments confirmed the indispensable role of the catalyst (entry 17).

### Substrate scope investigation

β-*O*-Aryl glycosides are valuable scaffolds in medicinal chemistry[43]. However, the selective synthesis of these compounds remains challenging because the reaction is under thermodynamic control, favoring the formation of *C*-aryl glycosides and leading to undesired rearrangement[44]. Using optimized conditions, we evaluated various phenolic nucleophiles with glycosyl carbamate donors (Fig. 3). Remarkably, the reactions proceeded with excellent stereocontrol, affording exclusively the 1,2-*trans O*-aryl glycosides irrespective of the

## Table 1 | Reaction optimization

| Entry | Catalyst | Solvent | Yield% | α/β ratio[a] |
|---|---|---|---|---|
| 1[b] | 20 mol% TMSOTf | CH₂Cl₂ | trace | -- |
| 2 | 20 mol% TfOH | CH₂Cl₂ | 53 | 3:1 |
| 3 | 20 mol% TsOH·H₂O | CH₂Cl₂ | 45 | 1:1 |
| 4[b] | 20 mol% BF₃·Et₂O | CH₂Cl₂ | 50 | 3:1 |
| 5 | 20 mol% SnCl₄ | CH₂Cl₂ | 52 | 1.6:1 |
| 6 | 20 mol% **A** | CH₂Cl₂ | 17 | 1:4.5 |
| 7 | 20 mol% **B** | CH₂Cl₂ | no reaction | -- |
| 8 | 20 mol% **C** | CH₂Cl₂ | no reaction | -- |
| 9 | 20 mol% **D** | CH₂Cl₂ | 71 | <1:20 |
| 10 | 20 mol% **E** | CH₂Cl₂ | 20 | 1:2.7 |
| 11 | 20 mol% **D** | CH₃CN or DMF | no reaction | -- |
| 12[c] | 5 mol% **D** | CH₂Cl₂ | 96 | <1:20 |
| 13[d] | 20 mol% **D** | CH₂Cl₂ | 13 | 1.7:1 |
| 14[e] | 5 mol% **D** | CH₂Cl₂ | 36 | 1.8:1 |
| 15[f] | 5 mol% **D** | CH₂Cl₂ | no reaction | -- |
| 16[g] | 5 mol% **D** | CH₂Cl₂ | no reaction | -- |
| 17 | none | CH₂Cl₂ | no reaction | -- |

Bn benzyl, Ac acetyl, Me methyl.

[a]Reaction conditions: **1a** (0.05 mmol, 1.0 equiv.), **2a** (0.055 mmol, 1.1 equiv.), catalyst (0.01 mmol, 20 mol%), CH₂Cl₂ (1 mL) at room temperature for 12 h, yields were determined by ¹⁹F NMR using trifluoromethoxybenzene as an internal standard; α/β ratio was determined by crude ¹H NMR spectrum.

[b]0 °C.

[c]**1a** (0.075 mmol), 93% of oxazolidinone was recovered.

[d]**1a-α** was used instead of **1a**.

[e]**1h** was used instead of **1a**, 40 °C.

[f]**1i** was used instead of **1a**.

[g]**1l** was used instead of **1a**.

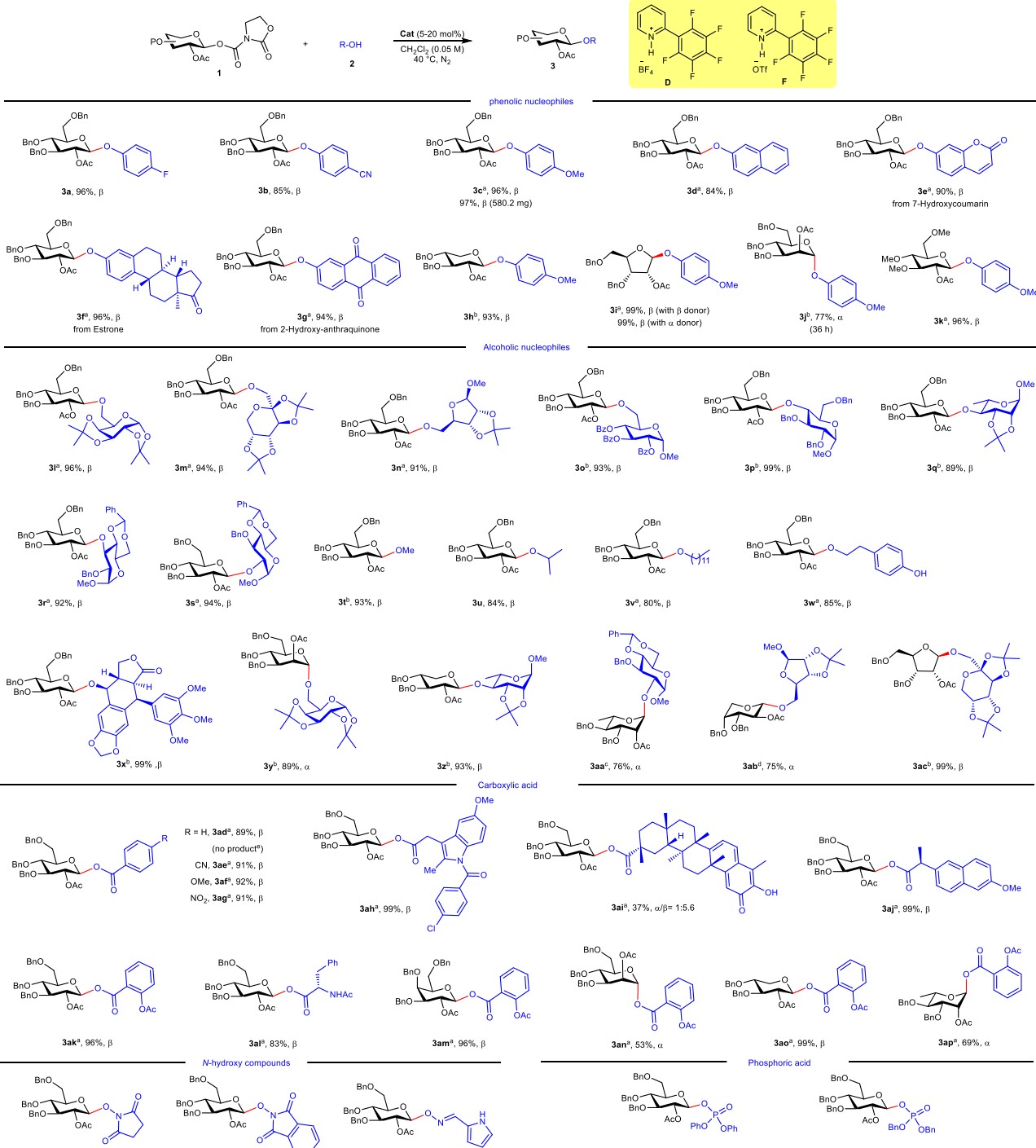

**Fig. 3 | Substrate scope.** Reaction condition: **1a** (0.075 mmol), **2a** (0.05 mmol), CH₂Cl₂ (1 mL), **D** (5 mol%), RT, 12 h, under N₂; [a]**D** (20 mol%), 40 °C; [b]**F** (20 mol%), 40 °C; [c]**F** (20 mol%), 50 °C, [d]**1a** (0.15 mmol), **F** (20 mol%), 40 °C; [e]36 h, no catalyst; [f]**F** (40 mol%). Isolated yields. Bn benzyl, Ac acetyl, Ph phenyl.

phenol electronic properties (**3a-3c**). Pleasingly, this method showed no scale-up effect. The reaction performed on a 1 mmol scale proceeded well, affording excellent yield and stereoselectivity (**3c**). Notably, β-naphthol, which typically favors *C*-glycosylation[44], exclusively formed the desired *O*-glycoside **3d**. The method also accommodated pharmaceutically relevant scaffolds including 2-hydroxycoumarin (**3e**), estrone (**3f**), and 2-hydroxylanthraquinone (**3g**). The scope was further extended to D-xylose, D-ribose, and D-mannose (**3h-3j**). While ribose-derived donors reacted efficiently under standard conditions, the xylosylation and mannosylation proceeded sluggishly with catalyst **D**. Notably, switching to triflate analog

**F** significantly enhanced the reaction, affording the α-mannoside (**3j**) in good yield. Intriguingly, the α-configured ribosyl donor also yielded exclusively the β-product (**3i**), underscoring neighboring group participation. Furthermore, this protocol was successfully applied to the Me-protected derivative, demonstrating the generality of the method (**3k**).

The glycosylation method also demonstrated good generality across diverse alcohol acceptors (Fig. 3), delivering glycosides (**3l-3v**) in high yields (85-99%) with excellent stereoselectivity. A wide range of carbohydrate acceptors reacted smoothly with donor **1a**. This versatility facilitated the preparation of β-(1→6)-, (1→4)-, (1→3)-, and

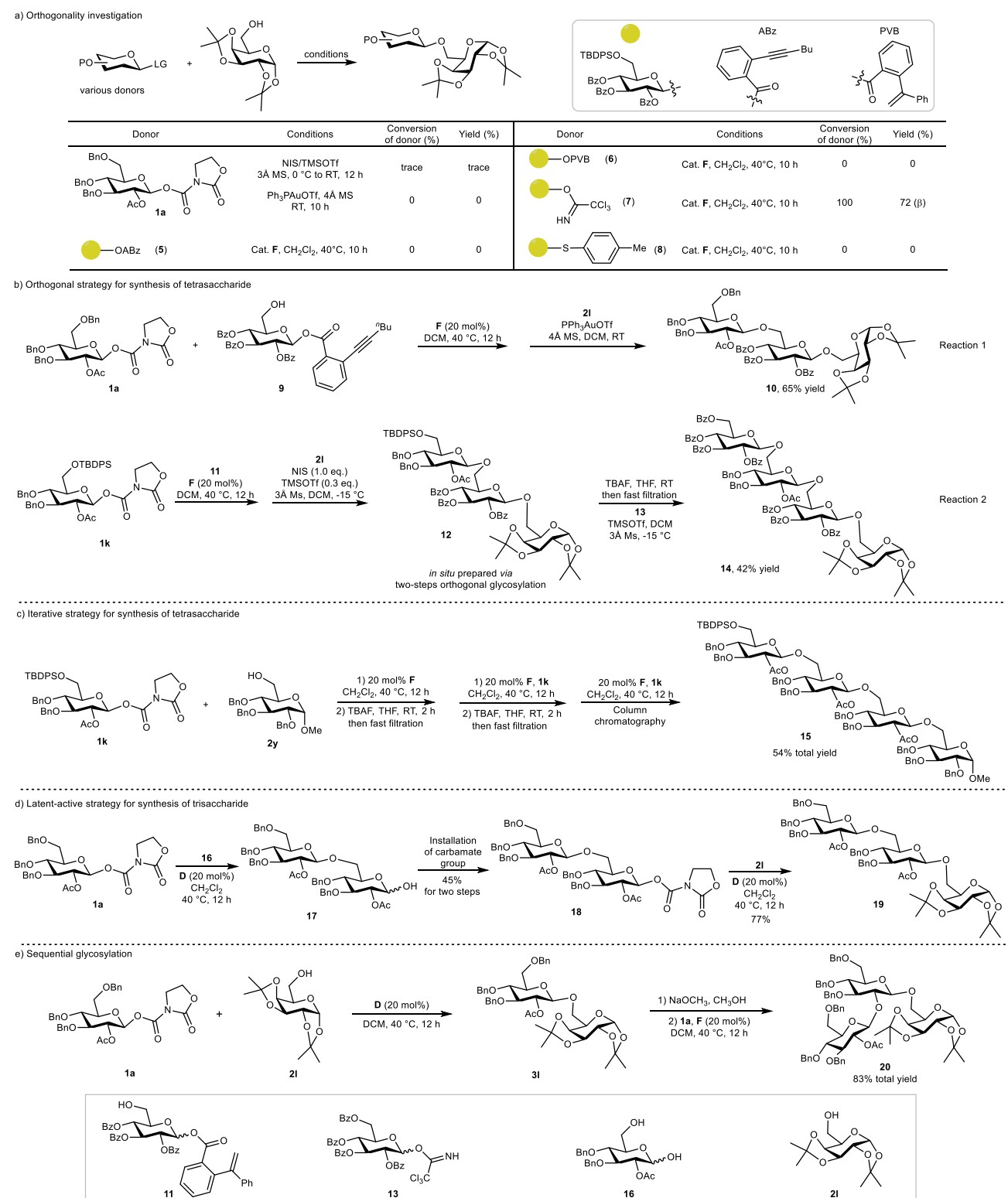

**Fig. 4 | Application of the method in oligosaccharide synthesis. a** Orthogonality investigation; **b** Orthogonal strategy for synthesis of tetrasaccharide; **c** Iterative strategy for synthesis of tetrasaccharide; **d** Latent-active strategy for synthesis of trisaccharide; **e** Sequential glycosylation. Isolated yields. ABz 2-(hexyn-1-yl)benzoyl, PVB 2-(1-phenylvinyl)benzoyl, � 2,3,4-tri-*O*-benzoyl-6-*O*-(*tert*-butyldiphenylsilyl)-D-glucosyl, Bn benzyl, Ac acetyl, Bz benzoyl, TBDPS *tert*-butyldiphenylsilyl, Bu butyl.

(1 → 2)-linked disaccharides (**3l-3s**) efficiently regardless of variations in protecting groups or hydroxyl position.

Aliphatic alcohols, ranging from small-chain substrates (methanol, **3t**; isopropanol, **3u**) to long-chain analogs (dodecanol, **3v**), were seamlessly converted. Specifically, selective glycosylation occurred preferentially at the aliphatic OH over phenolic OH in

4-hydroxyphenethyl alcohol (**3w**). The method was further validated by nearly quantitative glycosylation of the bioactive natural product podophyllotoxin (**3x**), a precursor to anticancer agents etoposide and teniposide[45]. The protocol demonstrated broad versatility, successfully converting multiple pyranosyl carbamate donors into their corresponding glycosides (**3y-3ac**) in 75–99% yields.

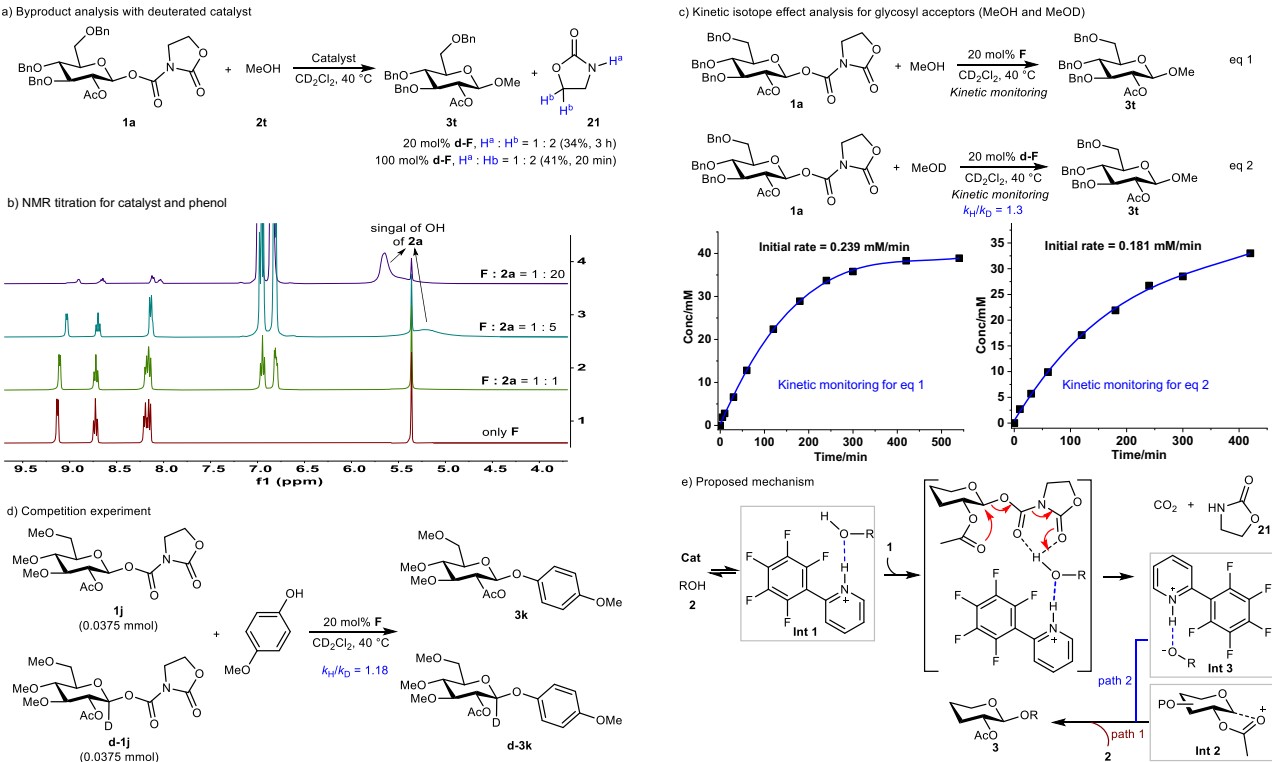

**Fig. 5 | Mechanistic studies. a** Byproduct analysis with deuterated catalyst (NMR yields); **b** NMR titration for catalyst and phenol; **c** Kinetic isotope effect analysis for glycosyl acceptors (MeOH and MeOD); **d** Competition experiment; **e** Proposed mechanism. Bn benzyl, Ac acetyl, Me methyl.

Remarkable scope and stereoselectivity were also demonstrated in the glycosylation of carboxylic acids[46] using this protocol. Both aromatic (benzoic acid derivatives, **3ad-3ag**) and pharmacologically relevant aliphatic molecules (indomethacin **3ah**, tripterine **3ai**, naproxen **3aj**, aspirin **3ak**), formed exclusive β-glycosyl esters. The protocol also accommodated amino acid derivative (acetyl-L-phenylalanine, **3al**) and various sugar configurations (**3am-3ap**). Control experiments confirmed that the pyridinium catalyst was indispensable, with no product formation occurring in its absence. The method further enabled efficient synthesis of aminooxy glycosides (**3aq-3as**) from *N*-hydroxysuccinimide, *N*-hydroxyphthalimide and oxime, providing streamlined access to biologically important aminooxy glycosides[47]. Intriguingly, phosphoric acid acceptors (diphenyl/dibenzyl hydrogen phosphates) reacted smoothly with donor **1a** to afford phosphate-linked glycosides[48] (**3at-3au**). It is noteworthy that the uncatalyzed reations exhibited significantly slower kinetics, underscoring the critical role of the catalyst in this transformation (**3at**).

### Application in oligosaccharide synthesis

Our oxazolidinone-based glycosyl carbamates enable oligosaccharide assembly through unique reactivity profiles distinct from established conditions. As shown in Fig. 4a, donor **1a** exhibited negligible reactivity under NIS/TMSOTf[24] conditions and remained inactive under gold-catalyzed conditions[22]. Among comparative donors (**5, 6, 7, 8**), only **7** participated in our pyridinium-catalyzed system. This excellent orthogonality allows for selective, stepwise oligosaccharide construction through strategic donor selection. Initially, we established orthogonal glycosylation between our donor **1a** and Yu's donor **9**, successfully obtaining trisaccharide **10** through a one-pot, two-step reaction (Fig. 4b, Reaction 1). Furthermore, a one-pot sequence involving glycosylation of donor **1k** and Xiao's donor **11**, followed by selective TBDPS deprotection and subsequent coupling with Schmidt's donor **13**, afforded tetrasaccharide **14** (Fig. 4b, Reaction 2). Then, we rapidly constructed tetrasaccharide **15** through three consecutive

pyridinium-catalyzed glycosylation steps employing an iterative strategy (Fig. 4c). Interestingly, latent-active strategy glycosylation proved successful (Fig. 4d). Donor **1a** reacted selectively with the hydroxy group at C6 position of **16**, leaving the anomeric hydroxy group untouched. Subsequent installation of carbamate moiety afforded **18**, which served as a competent donor for futher glycosylation with **2l** to produce trisaccharide **19**. Notably, the C2 acetyl group served as a useful handle for controlled assembly of oligosaccharide **20** through selective deprotection-glycosylation sequences (Fig. 4e).

### Mechanistic studies

To elucidate the reaction mechanism, we conducted a series of experiments. Initial attempt employing deuterated catalyst **d-F**, even with a stoichiometric amount, revealed no deuterium transfer to the oxazolidinone (Fig. 5a), ruling out the direct donor activation. We therefore proposed that the catalyst might initially bind the glycosyl acceptor to form an intermediate. In the NMR titration of catalyst **F** with phenol **2a**, an upfield proton shift of the catalyst and disappearance of the phenolic OH signal (Fig. 5b) indicated hydrogen bonding between the phenol -OH and catalyst -NH, in line with a previous report[40] and supporting initial catalyst-acceptor interaction. Subsequent kinetic studies using MeOD and MeOH demonstrated a primary kinetic isotope effect (KIE) of only 1.3, suggesting O-H bond cleavage is not the rate-determining step (Fig. 5c)[49]. Additionally, the 1,2-*trans* product configuration implied oxocarbenium involvement[50]. Parallel experiments with deuterated donor **d-1j** *vs* **1j** yielded a secondary KIE of 1.18 (Fig. 5d), supporting sp[3] to sp[2] rehybridization[51]. These findings collectively suggest that the oxocarbenium generation as the rate-determining step. Based on experimental evidence and literature precedents[50], a plausible mechanism is proposed in Fig. 5e. The reaction starts with the formation of **Int 1**, which subsequently activates the glycosyl donor **1**, leading to the generation of the oxocarbenium (**Int 2**), with concomitant release of $CO_2$ and oxazolidinone **21**. In this process, the hydroxyl proton of **Int 1** is proposed to interact with the carbonyl groups of the glycosyl donor **1** through

hydrogen bonding, while the neighboring participation effect of the 2-OAc group facilitates the departure of the anomeric leaving group, thereby promoting the efficient formation of the oxocarbenium **Int 2**. Concurrently, **Int 1** is converted into **Int 3**. Finally, **Int 2** is intercepted either by the glycosyl acceptor **2** (path 1) or **Int 3** (path 2) to afford the product **3**.

## Discussion

In conclusion, we have developed an efficient atom-economical glycosylation method for *O*-glycosides synthesis using stable glycosyl carbamates and cost-effective pyridinium salts as catalysts, with the $CO_2$ release and oxazolidinone recovery. This protocol exhibits broad applicability, accommodating diverse glycosyl donors and acceptors while enabling orthogonal, iterative, and latent–active strategies for oligosaccharide construction. Mechanistic studies suggest that the pyridinium catalyst initially interacts with the glycosyl acceptor to generate a reactive intermediate, promoting carbamate activation. Kinetic isotope effect analysis implicates oxocarbenium formation as the likely rate-determining step.

## Methods

### General procedure for pyridinium-catalyzed *O*-glycosylation

To an oven-dried vial was added glycosyl carbamate **1** (0.075 mmol, 1.5 equiv.), glycosyl acceptor **2** (0.05 mmol, 1.0 equiv.), catalyst **D** (5 mol%) and anhydrous $CH_2Cl_2$ (1 mL) under nitrogen atmosphere. The solution was stirred at room temperature for 12 h. The resulting mixture was concentrated and the residue was purified by silica gel column chromatography to afford the product **3**.

### Procedure for gram-scale reaction

In a glove box filled with nitrogen, to an oven-dried 25 mL tube equipped with a stirring bar were added **1a** (0.9078 g, 1.5 mmol, 1.5 equiv.), acceptor **2c** (124.2 mg, 1.0 mmol, 1.0 equiv.), **D** (66.0 mg, 20 mol%), and anhydrous $CH_2Cl_2$ (7 mL, 0.14 M). The reaction mixture was stirred at 40 °C for 12 h and then purified by column chromatography on silica gel with petroleum ether/ethyl acetate (5:1) as eluent to afford **3c** as a white solid (580.2 mg, 97% yield).

## Data availability

The authors declare that the data supporting the findings of this study are available within the article and its Supplementary Information Files. Additional data are available from the corresponding author upon request. The X-ray crystallographic coordinates for structures reported in this study have been deposited at the Cambridge Crystallographic Data Center (CCDC), under deposition numbers 2433567 (**1a**). These data can be obtained free of charge from The Cambridge Crystallographic Data Centre via www.ccdc.cam.ac.uk/data_request/cif.

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

## Acknowledgements

We appreciate the National Natural Science Foundation of China (22201041 (C.X.), 22208055 (S.W.)), the Fuzhou University (511041 (C.X.)), the Fujian Provincial Natural Science Foundation of China (2020J011002 (W.C.), 2025J01783 (T.L.)), and the Joint Funds for the Innovation of Science and Technology, Fujian Province (2023Y9140 (W.C.)) for financial support. We greatly thank Dr. Sebastian Hui for his valuable assistance in language editing.

## Author contributions

X.Q. conducted the majority of the experimental work. L.K., Q.J., W.Z., W.L., J.L. helped with expansion of substrate scope. W.C., T.L. and S.W. discussed the project. C.X. conceived the idea and supervised the project. C.X. and X.Q. prepared this manuscript. All authors contributed to data analysis and commented on the manuscript.

## Competing interests

The authors declare no competing interests.
