## [Transparent Peer Review file · Nature Communications]

A Universal O-Glycosylation Platform Enabled by Pyridinium Catalysis Using Gas-Releasing Oxazolidinone-Based Carbamates Donors

Corresponding Author: Professor Chunfa Xu

Version 0:

Reviewer comments:

Reviewer #1

(Remarks to the Author)

The manuscript titled "A Universal O-Glycosylation Platform Enabled by Pyridinium Catalysis Using Gas-Releasing Oxazolidinone-Based Carbamates Donors" by Xu and co-workers have reported the activation of bench stable Oxazolidinone-based carbamates as glycosyl donors using a novel pyridinium salt. The method provides a convenient route towards the β -glycosides where the stereoselectivity is attributed to the neighbouring group effect of the C2-acetate. Exclusive stereoselectivity has been observed with diverse range of nucleophiles that afforded varied glycosides from different donor configurations. The selective activation of the carbamate donors under the reaction conditions in comparison to the other well-known donors have also been employed for oligosaccharide synthesis. The work may be considered for publication after addressing the following concerns:

1. The authors report the presence of C2-OAc crucial for the observed transformation based on the reactions in Fig. 3. While the high β -selectivity observed for donor 1a is attributed to NGP by OAc as compared to the benzyl ether bearing donor 1h, the difference in reaction conversion for these donors is not explicit. The reaction mechanism proposed by the authors in Fig. 6e involves oxocarbenium ion formation. Suitable explanation for the low yield (36%) for the donor 1h in the reaction even at 40 °C must be provided and the mechanistic pathway must corroborate this observation. The effect of the C2-group in the reactivity of the substrate and formation of the oxocarbenium ion in the rate determining step must be established in the reaction mechanism.

2. The reactions have been performed on a very small scale of typically 0.075 mmol of the glycosyl donors. At least one scale up reaction up to 1 mmol must be performed to demonstrate the large-scale application of the method.

3. The anomeric peak for the synthesized glycosides must be identified for a few products by performing correlation spectra. NMR evidence for the high β selectivity must be included⁴

4. The authors must re-check the manuscript for several grammatical errors.

(i) Please re-write in correct language: "Glycosylation of carboxylic acids was demonstrated remarkable scope and stereoselectivity in this protocol."

(ii) Complete the sentence: "Notably, while uncatalyzed reactions proceeded slowly (3at)."

Reviewer #2

(Remarks to the Author)

In this manuscript, the authors described a pyridinium salt-catalyzed glycosylation reaction using gas-releasing oxazolidinone-based carbamate donors. The glycosyl donor and catalyst are new. The reaction is efficient. The glycosylation can be used for oligosaccharide synthesis. And the reaction mechanism was investigated. The disclosed protocol will provide alternative tools for oligosaccharide assembly. In my opinion, this manuscript is suitable for publication in this journal after some revisions.

1) The current mechanism only mentioned the binding between the pyridinium catalyst and glycosyl acceptor. As the authors say, the oxocarbenium formation is likely the rate-determining step. The authors should depict how the donor 1 convert to the oxocarbenium Int 2.

2) For compounds 3at and 3au, ³¹P NMR should be provided.

3) The manuscript requires significant proof reading and revision to improve the quality of English.

Version 1:

Reviewer comments:

Reviewer #1

(Remarks to the Author)

The authors have addressed all the comments raised by both the reviewers and have revised the manuscript as well as SI file satisfactorily. The manuscript in the present form may be accepted for publication. However, few spelling mistakes are still there which must be re-checked before final version being published.

Reviewer #2

(Remarks to the Author)

The authors addressed my concerns. Now the manuscript can be accepted for publication.

Point-to-point response

#Reviewer 1

Comment 1: *The manuscript titled “A Universal O-Glycosylation Platform Enabled by Pyridinium Catalysis Using Gas-Releasing Oxazolidinone-Based Carbamates Donors” by Xu and co-workers have reported the activation of bench stable Oxazolidinone-based carbamates as glycosyl donors using a novel pyridinium salt. The method provides a convenient route towards the β -glycosides where the stereoselectivity is attributed to the neighbouring group effect of the C2-acetate. Exclusive stereoselectivity has been observed with diverse range of nucleophiles that afforded varied glycosides from different donor configurations. The selective activation of the carbamate donors under the reaction conditions in comparison to the other well-known donors have also been employed for oligosaccharide synthesis. The work may be considered for publication after addressing the following concerns.*

Response: We sincerely thank the reviewer for the careful reading of our manuscript and for the constructive and encouraging comments. We have carefully addressed all the points raised in the review.

Comment 2: *The authors report the presence of C2-OAc crucial for the observed transformation based on the reactions in Fig. 3. While the high α -selectivity observed for donor **1a** is attributed to NGP by OAc as compared to the benzyl ether bearing donor **1h**, the difference in reaction conversion for these donors is not explicit. The reaction mechanism proposed by the authors in Fig. 6e involves oxocarbenium ion formation. Suitable explanation for the low yield (36%) for the donor **1h** in the reaction even at 40 °C must be provided and the mechanistic pathway must corroborate this observation. The effect of the C2-group in the reactivity of the substrate and formation of the oxocarbenium ion in the rate determining step must be established in the reaction mechanism.*

Response: We greatly thank the reviewer for this valuable comment. For the reaction involving the fully benzyl-protected glycosyl donor **1h**, only a 36% yield was obtained, with 55% of the starting material **1h** being recovered. The reactivity of **1h** was found to be significantly lower than that of the 2-OAc-protected donor **1a**. As reported by Demchenko *et al.* in *Journal of Organic Chemistry* (Superarming Common Glycosyl Donors by Simple 2-O-Benzoyl-3,4,6-tri-O-benzyl Protection. *J. Org. Chem.* **2009**, 75, 1095-1100), the 2-O-benzoyl group can facilitate the departure of the anomeric leaving group through a neighboring group assistance effect. Based on the Demchenko's report and our observed difference in reactivity between **1h** and **1a**, we hypothesize that a

similar assisting effect of the 2-OAc group may operate in our glycosylation system. To further verify this assumption, we synthesized the α -anomeric glycosyl donor **1a- α** . In this configuration, the 2-OAc group occupies an equatorial position, while the anomeric leaving group resides in an axial orientation. As a result, their spatial arrangement is not antiperiplanar, and thus the 2-OAc group can not effectively assist the departure of the anomeric leaving group. When subjected to the catalytic conditions, the reaction afforded only 13% yield even with 20 mol% of catalyst, which is consistent with our hypothesis. The reactivity of **1a- α** has been added to the reaction optimization table (Fig 3, entry 13). Moreover, the neighboring group participation of the 2-OAc substituent facilitating the departure of the anomeric leaving group has now been incorporated into the proposed reaction mechanism in the revised manuscript (Fig 6e).

Comment 3: *The reactions have been performed on a very small scale of typically 0.075 mmol of the glycosyl donors. At least one scale up reaction up to 1 mmol must be performed to demonstrate the large-scale application of the method.*

Response: We appreciate the reviewer's valuable suggestion. A scale-up experiment using 1 mmol of the glycosyl acceptor **2c** and 1.5 mmol of the donor **1a** was conducted, affording the desired product **3c** in 97% yield with β -selectivity. The reaction outcome indicates that scaling up has negligible impact on the efficiency or stereoselectivity of this methodology. The corresponding results have been added to the revised manuscript (page 2, right column, in "Substrate scope investigation" section) and SI (page 55).

Comment 4: *The anomeric peak for the synthesized glycosides must be identified for a few products by performing correlation spectra. NMR evidence for the high β selectivity must be included.*

Response: We sincerely thank the reviewer for pointing this out. In the original Supplementary Information file, we have conducted comprehensive 2D NMR analysis for the glycosyl donors and representative products, and the key proton signals used to determine the anomeric configurations have been clearly labeled in the NOESY spectrum. Specifically, spectra for mannosyl donor **1b** (Figures S34–S39), galactosyl donor **1c** (Figures S40–S45), xylosyl donor **1d** (Figures S46–S51), arabinosyl donor **1e** (Figures S52–S55), rhamnosyl donor **1f** (Figures S56–S60), and ribosyl donor **1g** (Figures S61–S66) were included. The corresponding products, such as xylose product **3h** (Figures S106–S111), ribose product **3i** (Figures S112–S117), mannose product **3y** (Figures S148–S153), rhamnose product **3z** (Figures S156–S161), arabinose product **3ab** (Figures S162–S167), and galactose product **3am** (Figures S188–S193), were also analyzed. For certain glucosyl derivatives, such as compound **3s**, the obtained NMR data are consistent with those reported in the literature (*J. Am. Chem. Soc.* **2020**, *142*, 7235–7242), further validating our stereochemical assignments.

In addition, representative NMR spectra demonstrating the high β -selectivity have now been included in the Supplementary Information (Pages S20–S23), and a summary of representative crude spectra is provided in Section 12 (Pages S176–S184).

Comment 5: *The authors must re-check the manuscript for several grammatical errors. (i) Please re-write in correct language: “Glycosylation of carboxylic acids was demonstrated remarkable scope and stereoselectivity in this protocol.” (ii) Complete the sentence: “Notably, while uncatalyzed reactions proceeded slowly (3at).”*

Response: We sincerely thank the reviewer for pointing out the grammatical errors in our manuscript. As non-native English speakers, we acknowledge that our initial submission contained several linguistic inaccuracies, and we apologize for any inconvenience this may have caused. During the revision process, we invited Dr. Sebastian Hui, a Hong Kong native who obtained his PhD in Organic Chemistry from the Technical University of Dortmund (Germany), to assist us in polishing the manuscript for language accuracy and clarity. All grammatical corrections have now been carefully implemented, and the revised portions are highlighted in yellow in the manuscript. In recognition of his contribution, we have also included an acknowledgment to Dr. Hui in the revised version.

#Reviewer 2

Comment 1: *In this manuscript, the authors described a pyridinium salt-catalyzed glycosylation reaction using gas-releasing oxazolidinone-based carbamate donors. The glycosyl donor and catalyst are new. The reaction is efficient. The glycosylation can be used for oligosaccharide synthesis. And the reaction mechanism was investigated. The disclosed protocol will provide alternative tools for oligosaccharide assembly. In my opinion, this manuscript is suitable for publication in this journal after some revisions.*

Response: We sincerely thank the reviewer for the positive and encouraging comments regarding our work.

Comment 2: *The current mechanism only mentioned the binding between the pyridinium catalyst and glycosyl acceptor. As the authors say, the oxocarbenium formation is likely the rate-determining step. The authors should depict how the donor 1 convert to the oxocarbenium Int 2.*

Response: We appreciate the reviewer's insightful comment. Experimental results showed that fully benzyl-protected donor **1h** and the **1a- α** anomer exhibited significantly lower reactivity than **1a**, indicating that the 2-OAc substituent possibly facilitates the departure of the anomeric leaving group of glycosyl donor **1a**. This phenomenon has also been reported in Demchenko's work (Superarming Common Glycosyl Donors by Simple 2-O-Benzoyl-3,4,6-tri-O-benzyl Protection. *J. Org. Chem.* **2009**, 75, 1095-1100). Therefore, we propose that **Int 1** interacts with the carbonyl groups of donor **1** through hydrogen bonding, leading to donor activation. Subsequently, with the neighboring group participation of the 2-OAc substituent, the anomeric leaving group departs to form the oxocarbenium intermediate **Int 2**. This proposed process has been incorporated into the revised manuscript.

Comment 3: For compounds **3at** and **3au**, ^{31}P NMR should be provided.

Response: We thank the reviewer for this valuable suggestion. The ^{31}P NMR spectra for compounds **3at** and **3au** have been provided in the Supplementary Information (Page 168, Figure S208. Page 169, Figure S211).

Comment 4: The manuscript requires significant proof reading and revision to improve the quality of English.

Response: We sincerely thank the reviewer for the valuable comment. We acknowledge that the manuscript required further proofreading and language refinement, as also noted by Reviewer 1. To address this issue, we invited Dr. Sebastian Hui, a Hong Kong researcher who obtained his Ph.D. in Organic Chemistry from the Technical University of Dortmund (Germany), to assist us in thoroughly revising and polishing the English throughout the manuscript. All corrections have been carefully implemented, and the revised text has been highlighted in yellow in the updated version.